# The impact of COVID-19 pandemic on cardiac rehabilitation of patients following acute coronary syndrome

**Feras Haskiah**[1], **Rana Jbara**[2], **Saar Minha**[3,4], **Abid Assali**[4,5], **Yaron Sela**[6], **David Pereg**[4,5]*

**1** Department of Internal Medicine D, Meir Medical Center, Kfar Saba, Israel, **2** Rappaport Faculty of Medicine, Technion Institute of Technology, Haifa, Israel, **3** Interventional Cardiology, Shamir Medical Center, Be'er-Yaakov, Israel, **4** Sackler Faculty of Medicine, Tel-Aviv University, Tel-Aviv, Israel, **5** Department of Cardiology, Meir Medical Center, Kfar Saba, Israel, **6** Baruch Ivcher School of Psychology, Reichman University, Herzliya, Israel

* davidpe@tauex.tau.ac.il

## Abstract

### Background

Cardiac rehabilitation improves prognosis and symptoms in cardiac patients. In 2020, due to the COVID-19 pandemic, cardiac rehabilitation services were temporarily suspended between April and August. We aimed to investigate the effect of cardiac rehabilitation suspension during the COVID-19 pandemic on patients' exercise capacity and metabolic parameters.

### Methods

Included were patients undergoing cardiac rehabilitation following hospital admission for ACS. Exercise capacity, weight and body fat percentage were compared between baseline, pre- and post-lockdown visits.

### Results

A total of 281 patients participated in the cardiac rehabilitation program prior to its suspension. Of them, only 198 (70%) patients returned to the program on its renewal and were included in the analysis. Exercise capacity improved significantly in the pre-lockdown stress test compared to baseline. However, there was a significant decrease in exercise capacity in the post compared to pre-lockdown test (8.1±6.3 and 7.1±2.1 METs in pre- and post-lockdown measurements, respectively, p<0.001). Of the 99 (50%) of patients that demonstrated at least 10% improvement in exercise capacity in the pre-lockdown test, 48 (48.5%) patients returned to their baseline values in the post-lockdown test. Post-lockdown assessment demonstrated a significant weight gain (80.3 and 81.1kg, in pre- and post-lock-down measurements, respectively, p<0.001) as well as an increase in visceral fat level and body fat percentage.

**Data Availability Statement:** All relevant data are within the paper and its Supporting information files.

**Funding:** The author(s) received no specific funding for this work.

## Conclusions

Cardiac rehabilitation suspension for 4 months during COVID-19 pandemic caused a significant reduction in exercise capacity and increased weight and body fat percent. These findings highlight the importance of remote cardiac rehabilitation services that can continue uninterrupted in times of pandemic.

## Introduction

Exercise based cardiac rehabilitation improves prognosis and quality of life of patients with coronary artery disease [1–3]. Accordingly, it is strongly recommended by all international guidelines for patients following hospitalization for acute coronary syndrome [4–7]. The main goals of cardiac rehabilitation programs are adequate risk factor control, optimizing lifestyle and improving exercise capacity [3]. Regular uninterrupted attendance of patients is very important in order to achieve maximal improvement in exercise capacity and better clinical outcomes [8, 9].

On January 30, 2020, the World Health Organization announced the outbreak of a new Corona virus. Soon after, the COVID-19 outbreak in China progressed into a worldwide pandemic [10, 11]. First cases of confirmed COVID-19 infections were reported in Israel in February 2020 and subsequently the Israeli government and health authorities implemented an escalating social distancing policy that led to several successive lockdowns. Moreover, the official recommendation for patients at high risk for severe COVID-19 infection (including elderly patients and those with cardiovascular comorbidities) was to remain in quarantine even between lockdowns [12, 13]. As a result, cardiac rehabilitation services were temporarily suspended between April and August 2020.

The aim of our study was to evaluate the effect of cardiac rehabilitation suspension for 4 months during the Covid-19 pandemic on the exercise capacity of patients participating in a cardiac rehabilitation program.

## Methods

This is an observational retrospective study that was conducted in the Cardiac Rehabilitation Center at the Meir Medical Center. Included were patients with a history of hospital admission due to acute coronary syndrome (ACS) who had been participating in the cardiac rehabilitation program for at least 3 months prior to the program suspension in April 2020.

A cardiac rehabilitation program for patients with acute coronary syndrome usually begins 4–6 weeks following hospital discharge and all costs are fully covered by the governmental medical insurance for 9 months. The exercise program consists of two outpatient sessions per week and a customized physical training program that is tailored for each patient under the guidance and supervision of a physiologist and a large staff of physical instructors. Each exercise session starts with a 5-minute warm-up, lasts 60 minutes and ends with a 5-minute cooling and stretching period. Exercise modalities vary between upper body ergometer, cycle ergometer, treadmills, stair climbers and light resistance exercises. During exercise sessions different modalities are combined according to indication and contra-indications established by biomechanical evaluation, adherence level to physical activity and musculoskeletal comorbidities. For each patient, physical activity intensity level is controlled and adjusted with the help of the Borg Rating of Perceived Exertion Scale (RPE scale; numbered from 6 to 20) [14]. Besides

physical training, the cardiac rehabilitation program includes psychosocial support, nutritional and lifestyle consultations. The cardiac rehabilitation program includes a complete assessment by a cardiologist at baseline and then every 3 months. This includes clinical assessment, the evaluation of exercise capacity and the measurement of weight, body fat percentage, and visceral fat level.

Exercise capacity was evaluated by a routine treadmill stress test using either Bruce or low-fit Bruce exercise protocols at baseline and then routinely every 3 months. Results from stress tests were documented by the supervising exercise physiologist or physician. Stress tests and metabolic equivalent (METs) evaluation were performed according to the American Heart Association Standards for Testing and Training [15, 16].

Weight, body fat percentage and visceral fat level were measured at baseline, and then routinely every 3 months using the Omron Karada Scan HBF511 (Omron Health Care, Kyoto, Japan) bioimpedance device (BIA) [17]. BIA is a noninvasive method for body composition assessment. After entering age, gender and height, patients were asked to stand on the device after its calibration. The measurements were held before the exercise sessions in order to avoid dehydration that may affect the accuracy of this method. The BIA measured the visceral fat level in the range from 1 to 59 (low to high level) [18, 19]. A higher level indicated more visceral fat.

The primary endpoint of our study was the rate of at least 10 or 25% improvement in exercise capacity in pre and post lockdown exercise test compared to baseline [20]. Secondary endpoints included the effect of cardiac rehabilitation suspension on weight, body fat percentage, and visceral fat level as well as LDL-cholesterol levels.

Clinical, laboratory and demographic data were extracted from the electronic medical records of Meir Medical center.

The study was approved by the Meir Medical center Institutional Ethics Committee in keeping with the principles of the Declaration of Helsinki. In accordance with Ministry of Health regulations, the Institutional Ethics Committee did not require written informed consent because data were collected anonymously from the electronic medical records without active patient participation.

## Statistical analysis

Descriptive statistics were performed using means and standard deviations for the continuous variables, and frequencies for the discrete variables. Univariate comparisons for repeated measures were performed using Friedman and Wilcoxon signed ranks test. Tests between independent samples were done using the Mann-Whitney test. Significance was considered for a p-value lower than 5%. Data were analyzed using SPSS software version 25.

## Results

A total of 281 patients participated in the cardiac rehabilitation program for at least 3 months prior to its suspension in April 2020. Of them, only 198 (70%) patients returned to the program on its renewal on August 2020 and were included in the analysis. In 75 of the 83 patients (90%) who did not return to cardiac rehabilitation at its renewal, the reason was patients' preference to maintain social distancing due to their concerns regarding Covid-19 contagion. Participants had a median age of 67.9±9.8 years and included 65 (32.8%) women. Table 1 presents baseline demographic and clinical characteristics of the study participants at the first cardiac rehabilitation appointment.

Exercise capacity improved significantly in the pre-lockdown stress test compared to baseline (6.9±2.1 and 8.1±6.3 METs, p<0.001) (Fig 1). Accordingly, at least 10 or 25%

**Table 1. Baseline patient characteristics and clinical factors.**

| Demographic Characteristics | N = 198 |
| --- | --- |
| Sex | |
| • Male | 133(67.2%) |
| Age (years) | 67.9±9.8 |
| • < 65 | 67(33.8%) |
| • ≥ 65 | 131(66.2%) |
| **Baseline Measurements** | |
| Height (cm) | 169.7±9.3 |
| Weight (kg) | 81.5±15.7 |
| BMI (kg/m$^2$) | 28.2±4.5 |
| BMI ≥30 | 63(31.8%) |
| Body fat percentage (%) | 30.5±8.8 |
| Visceral fat level | 12.9±4.7 |
| **Clinical Characteristics** | |
| Dyslipidemia | 152(76.8%) |
| Diabetes mellitus | 72(36.3%) |
| Cigarette smoking | 55(27.8%) |
| Family history of CAD | 61(30.8%) |
| Atrial fibrillation | 29(14.6%) |
| PVD | 15(7.5%) |
| Heart Failure | 31(15.6%) |
| Systolic Function: | |
| • Normal | 113(57.1%) |
| • Mild dysfunction | 28(14.1%) |
| • Moderate dysfunction | 36(18.2%) |
| • Severe dysfunction | 5(2.5%) |
| PCI | 166(83.8%) |
| CABG | 43(21.7%) |
| **Medication** | |
| Aspirin | 157(79.2%) |
| P2Y12 inhibitors | 85(42.9%) |
| Statins | 166(83.8%) |
| Ezetimibe | 50(25.3%) |
| PCSK9 inhibitors | 8(4%) |
| **Blood Tests at baseline** | |
| Total Cholesterol (mg/dl) | 147.5±42 |
| Triglyceride (mg/dl) | 135.4±68 |
| LDL-C(mg/dl) | 76.2±31 |
| HDL-C (mg/dl) | 44.3±11.4 |
| HbA1C (%) | 6.2±0.9 |
| Hemoglobin (g/dl) | 13.8±1.5 |
| Creatinine (mg/dl) | 0.98±0.2 |

Values are mean ± standard deviation or N (%).

BMI-body mass index; CAD-coronary artery disease; PVD-peripheral vascular disease; PCI-percutaneous coronary intervention; CABG-Coronary artery bypass graft surgery; PCSK9- Proprotein convertase subtilisin/kexin type 9; LDL-C-low density lipoprotein cholesterol; HDL-C-high density lipoprotein cholesterol; HbA1C-HemiglobinA1C

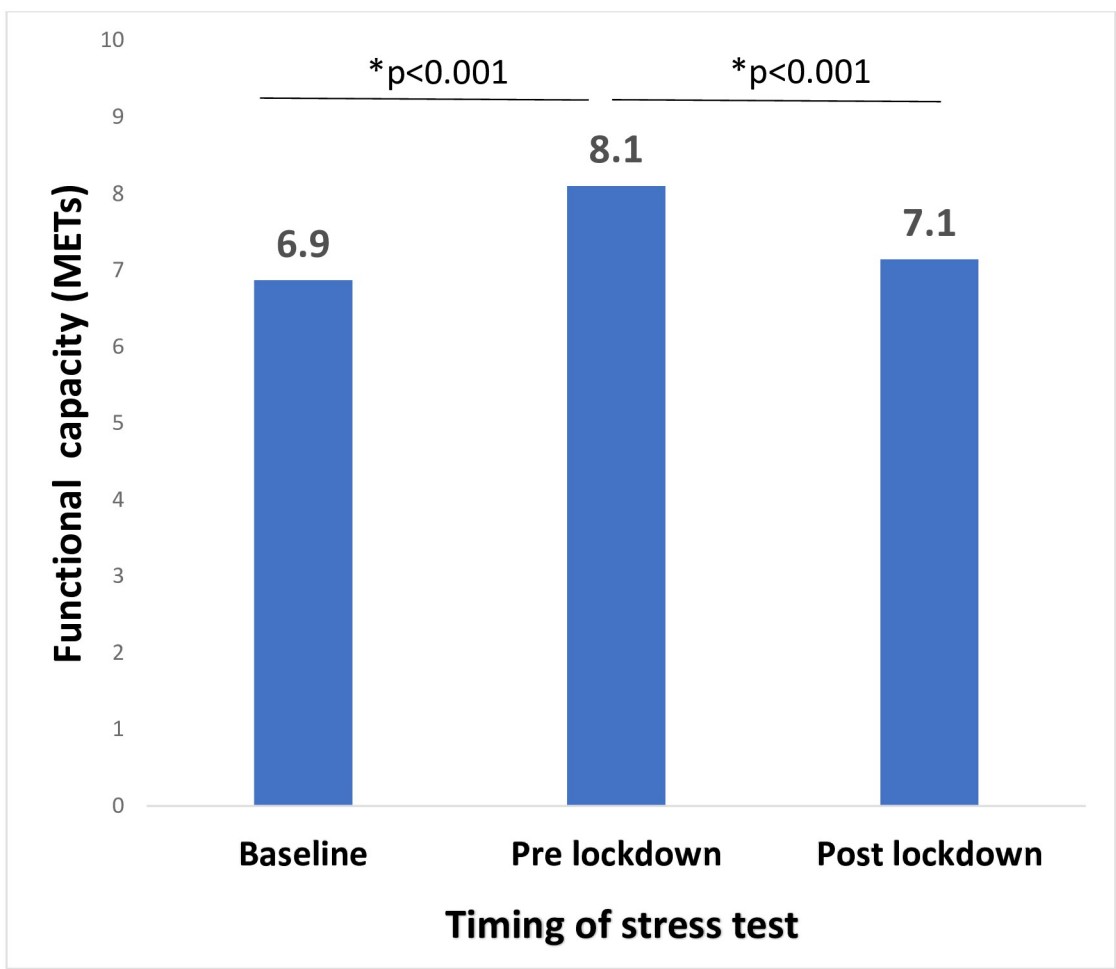

**Fig 1. Exercise capacity at baseline, pre-lockdown, and post lock down measurements (n = 198).**

improvement in exercise capacity was achieved by 50% and 17.7% of patients in the pre-lockdown compared to the baseline stress test. However, stress test on the resumption of cardiac rehabilitation following the end of the lockdown demonstrated a significant decrease in exercise capacity compared to the last pre-lockdown test (8.1±6.3 and 7.1±2.1 METs in pre- and post-lockdown measurements, respectively, p<0.001). Among the 99 (50%) patients who achieved at least 10% improvement in exercise capacity in the pre-lockdown stress test, only 48(48.5%) maintained this improvement in the post-lockdown stress test. Similar findings were demonstrated in the 17.7% patients who achieved at least 25% improvement in exercise capacity in the pre-lockdown stress test. This improvement was maintained by only '13 (37%) patients (Fig 2).

Participation in the cardiac rehabilitation program resulted in a significant mean weight reduction (81.5kg and 80.3kg in baseline and pre-lockdown measurement respectively, *p*<0.001) (Table 2). However, cardiac rehabilitation suspension resulted in a significant weight gain (80.3kg vs 81.1kg, in pre- and post-lockdown measurements respectively, p<0.001). Similar patterns were demonstrated for body fat percentage and visceral fat level (Table 2). Cardiac rehabilitation suspension had no effect on patients' lipid profile. Not surprisingly, albeit not statistically significant, LDL-cholesterol reduction was demonstrated in the pre-lockdown test

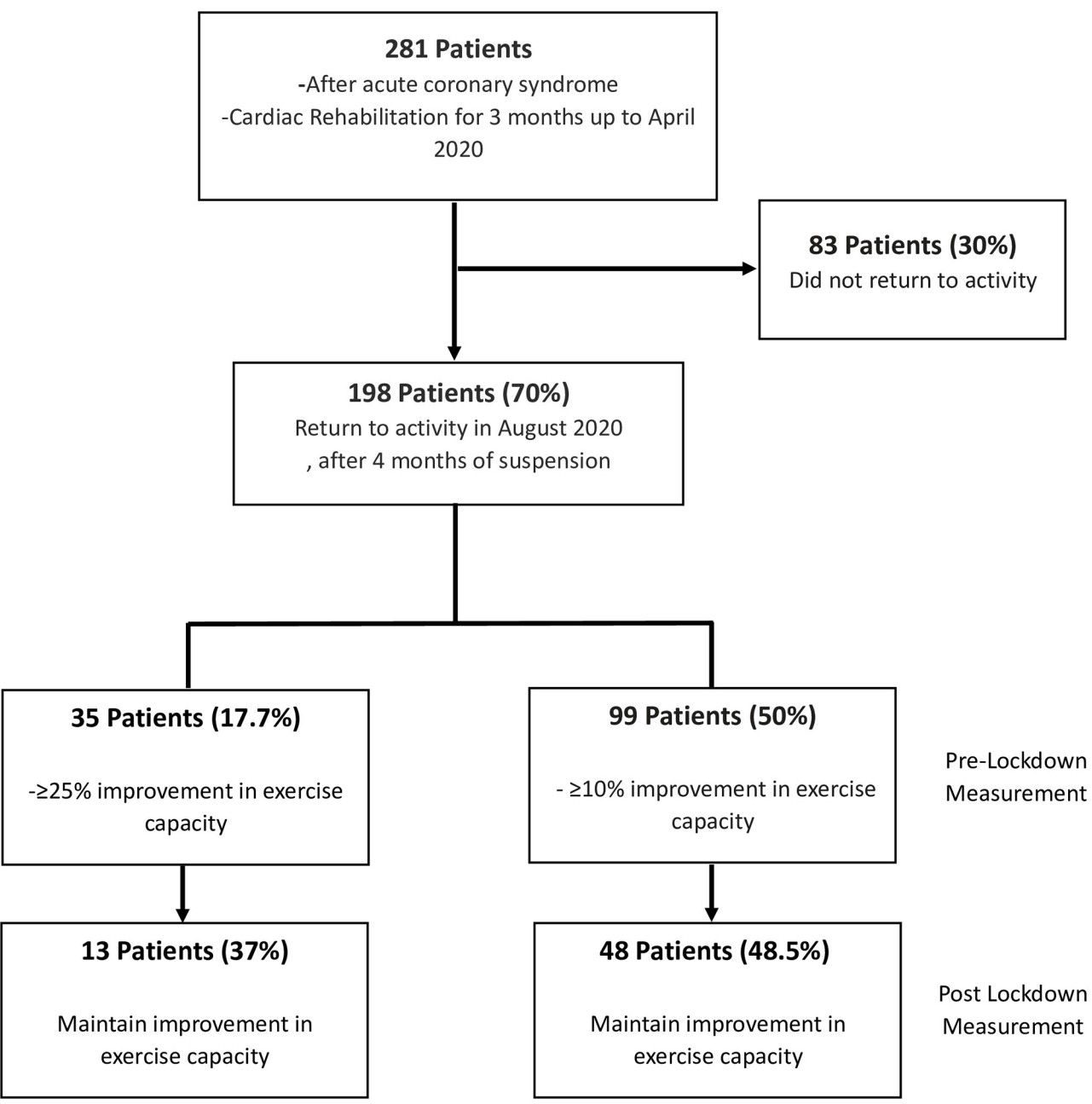

**Fig 2. Flow chart of the study participants including changes in exercise capacity (n = 198).**

followed by further consistent reduction during cardiac rehabilitation suspension (Table 2), suggesting adequate adherence to lipid lowering therapy during this period.

## Discussion

The current study investigated the effect of cardiac rehabilitation suspension during the COVID-19 pandemic on exercise capacity and metabolic parameters of the participants. We demonstrated that 30%(n = 83) of patients did not return to the cardiac rehabilitation at its

**Table 2. Secondary endpoints at baseline, pre-lockdown, and post lock down measurements.**

| Secondary Endpoint | Baseline | Pre-lock down | Post-lock down | P-value |
|---|---|---|---|---|
| Body Weight (kg) | 81.5±15.7 | 80.3±15.3 | 81.1±15.7 | <0.001* |
| BMI (kg/m²) | 28.2±4.5 | 27.7±4.3 | 28.2±4.6 | <0.001* |
| Body Fat Percentage (%) | 30.5±8.8 | 29.5±8.8 | 30.1±8.9 | <0.001* |
| Visceral Fat Level | 12.9±4.7 | 12.3±4.5 | 12.7±4.7 | <0.001* |
| LDL-C (mg/dl) | 76.2±31 | 72.1±30.1 | 67.8±30 | 0.35* |

Values are mean± SD

BMI-body mass Index; LDL-C -low-density lipoprotein c

*P-values are both for the comparison between baseline and pre-lockdown measurements and between pre and post lockdown measurements

renewal. Among patients who did return, the 4-month suspension was associated with a significant reduction of exercise capacity, a significant weight gain and increased body fat percentage and visceral fat level. On the other hand, cardiac rehabilitation suspension had no effect on LDL-cholesterol levels.

Exercise based cardiac rehabilitation improves outcomes and quality of life of patients following hospitalization for acute coronary syndrome [1–3]. Patients' adherence to cardiac rehabilitation programs has been associated with a lower risk of adverse outcomes and improved exercise capacity. A recent large meta-analysis demonstrated the dose-response association between cardiac rehabilitation session attendance or dose and reduced risk of major adverse cardiovascular events including death [21]. A continuous increase of 1 session in an uninterrupted cardiac rehabilitation program was significantly associated with a 1% to 2% reduction in major adverse cardiovascular event risk. Lack of adherence to cardiac rehabilitation may have several causes, including low patient compliance and concurrent acute cardiac and noncardiac conditions. The COVID-19 pandemic has caused a major disruption to the delivery of routine health care across the world. Cardiac rehabilitation programs across Israel have temporarily suspended in-person services as a result of large-scale physical distancing recommendations designed to flatten the COVID-19 pandemic curve. We demonstrated that a 4-month cardiac rehabilitation suspension eliminated most of the improvement in exercise capacity and metabolic parameters that had been achieved in the preceding months. Interestingly, we observed a consistent reduction in LDL-cholesterol levels during cardiac rehabilitation suspension, suggesting adequate adherence to lipid lowering therapy during this period. To the best of our knowledge, the current study is the first report on the effect of cardiac rehabilitation suspension during COVID-19 pandemic on patients with coronary artery disease.

Several studies have demonstrated that the lockdown and social distancing caused by the worldwide COVID-19 pandemic may influence common health behaviors including decreased daily physical activity and increased sedentary time [22, 23], unhealthy diet and increased body mass and obesity [24, 25]. These patterns have been demonstrated in different populations, including in patients with cardiovascular diseases [26]. Accordingly, it has been recommended to consider different strategies to allow home based physical activity programs [27, 28]. In recent years there has been a growing worldwide interest in remote cardiac rehabilitation programs using various technologies. Remote cardiac rehabilitation was found to be an effective, cost-efficient alternative delivery model that could function as a complement to existing services, improve overall utilization rates by increasing reach and satisfying unique participant preferences [29, 30]. Indeed, remote cardiac rehabilitation has been suggested as an effective alternative to outpatient cardiac rehabilitation in the COVID-19 pandemic era [31]. Our findings highlight the importance of remote cardiac rehabilitation especially in these days

of social restrictions. Recently, we have successfully initiated such a program in our medical center.

The current study has several limitations that merit consideration. First, similar to most studies on patients in cardiac rehabilitation, there is an inherent selection bias since patients who did not attend cardiac rehabilitation for various reasons (poor compliance, orthopedic comorbidities, social issues etc.) were not represented in the analysis. Second, exercise capacity was assessed by a regular treadmill stress testing, and we did not include a metabolic test (VO2 max) or evaluate muscle or vascular function. Third, we did not have data regarding the level of physical activity and dietary habits during cardiac rehabilitation suspension. Finally, we present the experience of a single cardiac rehabilitation center, and the cohort size is relatively modest. Therefore, our findings should be extrapolated to other populations with caution.

In conclusion, cardiac rehabilitation suspension for 4 months during COVID-19 pandemic was associated with a significant dropout of patients and reduction in exercise capacity and increased body mass and fat percent in the remaining participants. These findings highlight the importance of remote cardiac rehabilitation services that can continue uninterrupted in times of pandemic.

## Supporting information

**S1 Data.**
(XLSX)

## Author Contributions

**Conceptualization:** Feras Haskiah, David Pereg.

**Data curation:** Feras Haskiah, Rana Jbara, Saar Minha.

**Formal analysis:** Feras Haskiah, Yaron Sela.

**Investigation:** Abid Assali, David Pereg.

**Methodology:** Feras Haskiah, Saar Minha, Yaron Sela, David Pereg.

**Resources:** Feras Haskiah.

**Supervision:** Abid Assali, David Pereg.

**Validation:** Feras Haskiah.

**Visualization:** Feras Haskiah, Abid Assali.

**Writing – original draft:** Feras Haskiah, Saar Minha, David Pereg.

**Writing – review & editing:** Feras Haskiah, Rana Jbara, Saar Minha, David Pereg.

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
