## [Decision Letter · Decision Letter 0]

23 Jun 2022

PONE-D-22-14087The Impact of Covid-19 Pandemic on Cardiac Rehabilitation of patients following acute coronary syndrome.PLOS ONE

Dear Dr. Pereg,

Thank you for submitting your manuscript to PLOS ONE. After careful consideration, we feel that it has merit but does not fully meet PLOS ONE’s publication criteria as it currently stands. Therefore, we invite you to submit a revised version of the manuscript that addresses the points raised during the review process.

We look forward to receiving your revised manuscript.

Kind regards,

Giuseppe Limongelli

Academic Editor

PLOS ONE

Journal Requirements:

Reviewers' comments:

Reviewer's Responses to Questions

**Comments to the Author**

1. Is the manuscript technically sound, and do the data support the conclusions?

Reviewer #1: Partly

Reviewer #2: Yes

2. Has the statistical analysis been performed appropriately and rigorously? 

Reviewer #1: No

Reviewer #2: Yes

3. Have the authors made all data underlying the findings in their manuscript fully available?

Reviewer #1: Yes

Reviewer #2: Yes

4. Is the manuscript presented in an intelligible fashion and written in standard English?

Reviewer #1: Yes

Reviewer #2: Yes

5. Review Comments to the Author

Reviewer #1: 1) While the paper is within the word limit, I feel that readability would be enhanced by being more succinct and shortening the word count.

2) The authors should provide a more specific definition of the primary endpoint. The present definition ("The primary endpoint was the effect of cardiac rehabilitation suspension during COVID-19 pandemic on exercise capacity of the participants") appears too general.

3) The authors report that "tests between independent samples" were used. Independent sample tests are used to compare independent groups. Based on the study design, this approach does not seem correct for the analysis.

4) The sample size is relatively modest to offer robust evidence on the topic.

5) The authors should provide detailed information regarding all the reasons for not returning to cardiac rehabilitation.

6) The study population consisted of patients with previous acute coronary syndrome. Yet, only 79% of patients received aspirin. What were the reasons for not prescribing aspirin in the remaining 21% of cases? The same reasoning also apply to the prescription of statins.

7) The manuscript presentation, including the graphical presentation, should be substantially improved.

Reviewer #2: The manuscript by Prof. David Pereg and Colleagues analysed the impact of the COVID-19 pandemic on a cardiac rehabilitation program of coronary patients.

Briefly, the Authors found that cardiac rehabilitation (CR) suspension for 4 months during the COVID-19 pandemic caused a significant reduction in exercise capacity and increased weight and body fat percent highlighting the unmet need for specific remote measures to ensure better follow up strategies.

Currently, direct evidence on the impact of COVID-19 on CV rehabilitation programs is still limited and, for this reason the study is welcome. Although concise and clearly written, one may wonder if a more in-depth analysis on the present data could be performed to identify those patients at highest risk of losing CR benefit.

Please find my comments below.

Major Comments.

1. A total of 198/281 patients who were initially referred for CR returned for program continuation. Of the 99 patients who had achieved >10% improvement in exercise capacity, only 48 maintained this improvement. It would be interesting to acknowledge which factors were associated with lack of improvement (age, gender, BMI, baseline METs, LVEF, social factors?). A focus on this group could be helpful for the attending physician or nurse for future counselling. A logistic regression model could be a simple tool to assess this topic.

2. It is sometimes difficult to understand how the final population was derived and how many improved exercise capacity. A flowchart of the study population (even as supplementary material) could be useful.

3. In lines 157-159, Authors briefly refer to gender and age specific differences, however no quantitative data is reported. My suggestion is to either add data in the tables or remove the information altogether.

4. LDL concentrations seem to decrease through time although Authors highlight the lack of statistical significance. What test did they use? Was it a repeated measure test? By looking at the numbers the trend may seem significant.

5. A clearer presentation of the baseline vs. pre-lockdown vs. post lock-down characteristics of the study population could be useful: restructuring Table 2 to include trends of variables like body weight, BMI (and BMI >30), METs, etc. would be informative and may identify potential topics which could be amenable of future improvement.

Minor comments.

1. Introduction: The introduction paragraph has too many references (20).

2. Methods: the study design has not been reported.

3. Methods: a reference for the 10% cut-off to consider capacity improvement could be useful.

4. Tables: please add measurement units whenever missing and at least 1 decimal.

5. Table 1: how many patients were obese?

6. Table 1: do the Authors have echocardiographic information?

7. Table 1: PVD abbreviation is for PAD?

8. Figure 1: please add number of patients.

6. PLOS authors have the option to publish the peer review history of their article (what does this mean?). If published, this will include your full peer review and any attached files.

Reviewer #1: No

Reviewer #2: **Yes: **Carlo Fumagalli, MD

---

## [Author Response · Author response to Decision Letter 0]

19 Jul 2022

July,15,2022

Giuseppe Limongelli

Academic Editor

PLOS ONE

Ms. Ref. No.: PONE-D-22-14087

The Impact of Covid-19 Pandemic on Cardiac Rehabilitation of patients following acute coronary syndrome.

We thank the reviewers and you for the thorough review of our manuscript. We read the comments carefully and have revised the manuscript accordingly. Our responses to the comments are detailed below. We hope that you will find the revised manuscript suitable for publication in PLOS ONE. 

.

Sincerely yours,

David Pereg, MD 

Prof David Pereg, CICU director, Division of Cardiology, Meir Medical Center.

 59 Tchernichovsky St, Kfar-Saba, 44281, Israel.

 Tel: +972-9-7472587, Fax- +972-9-7472812. Email: davidpe@tauex.tau.ac.il

Response to Reviewer #1

Comment #1:

 While the paper is within the word limit, I feel that readability would be enhanced by being more succinct and shortening the word count.

Our response: Done.

Comment #2:

The authors should provide a more specific definition of the primary endpoint. The present definition ("The primary endpoint was the effect of cardiac rehabilitation suspension during COVID-19 pandemic on exercise capacity of the participants") appears too general.

Our response: Done. See lines 106-107 in the revised manuscript:

"The primary endpoint of our study was the rate of at least 10 or 25% improvement in exercise capacity in pre and post lockdown exercise test compared to baseline. "

Comment #3:

 The authors report that "tests between independent samples" were used. Independent sample tests are used to compare independent groups. Based on the study design, this approach does not seem correct for the analysis.

Our response:

Indeed, most of the primary analyses are dependent samples, and therefore we conducted Friedman and Wilcoxon signed ranks test. However, several comparisons were made between independent groups, such as males vs. females. We refer also to these comparisons.

Comment #4:

The sample size is relatively modest to offer robust evidence on the topic.

Our response:

We completely agree with this comment. The "study limitation" paragraph has been revised accordingly. Discussion section line 229-231: " Finally, we present the experience of a single cardiac rehabilitation center and the cohort size is relatively modest. Therefore, our findings should be extrapolated to other populations with caution."

Comment #5:

The authors should provide detailed information regarding all the reasons for not returning to cardiac rehabilitation.

Our response:

This comment is very well taken. Overall, 83 (30%) patients did not return to cardiac rehabilitation at its renewal. In 75 patients (90%) the reason for not returning to cardiac rehabilitation was patients' preference to maintain social distancing due to their concerns regarding Covid-19 contagion. This important information has been added to the results section of the revised manuscript. 

Result section, lines 128-130: " In 75 of the 83 patients (90%) who did not return to cardiac rehabilitation at its renewal, the reason was patients' preference to maintain social distancing due to their concerns regarding Covid-19 contagion. "

Comment#6:

The study population consisted of patients with previous acute coronary syndrome. Yet, only 79% of patients received aspirin. What were the reasons for not prescribing aspirin in the remaining 21% of cases? The same reasoning also apply to the prescription of statins.

Our response:

This comment is very well taken. Table 1 presents baseline medical therapy that was reported by the patients at the initiation of cardiac rehabilitation. In some cases, changes in therapy were made by the cardiac rehabilitation team during the rehabilitation period. The study population consisted of patients following hospital admission due to ACS. While aspirin is an integral part of the treatment of patients with ACS, there are several exceptions including patients with an indication for anticoagulation (atrial fibrillation, apical thrombus etc.) who are usually treated with a combination of an anti-coagulant and either aspirin or clopidogrel). Indeed, 15% of our study population had concomitant atrial fibrillation. Another group of patients who may not be treated with aspirin include those with MI with normal coronary arteries (MINOCA) who may be discharged with a single antiplatelet therapy. Regarding lipid lowering therapy, 88% of patients were treated with either statin or PCSK-9 inhibitors prior to initiation of cardiac rehabilitation. 

Comment #7:

The manuscript presentation, including the graphical presentation, should be substantially improved.

Our response: Done.

Response to Reviewer #2

Comment#1:

 A total of 198/281 patients who were initially referred for CR returned for program continuation. Of the 99 patients who had achieved >10% improvement in exercise capacity, only 48 maintained this improvement. It would be interesting to acknowledge which factors were associated with lack of improvement (age, gender, BMI, baseline METs, LVEF, social factors?). A focus on this group could be helpful for the attending physician or nurse for future counselling. A logistic regression model could be a simple tool to assess this topic

Our response:

We thank the reviewer for addressing this important point. Accordingly, we have performed a logistic regression model. However, none of the baseline characteristics was found to be an independent predictor for lack of improvement. 

Comment#2:

It is sometimes difficult to understand how the final population was derived and how many improved exercise capacity. A flowchart of the study population (even as supplementary material) could be useful.

Answer#2:

We thank the reviewer for this comment. A flowchart has been added to the revised manuscript accordingly- see figure 2 in the revised manuscript. 

Comment#3:

In lines 157-159, Authors briefly refer to gender and age specific differences, however no quantitative data is reported. My suggestion is to either add data in the tables or remove the information altogether.

Our response: Done. The relevant sentence has been removed. 

Comment#4:

LDL concentrations seem to decrease through time although Authors highlight the lack of statistical significance. What test did they use? Was it a repeated measure test? By looking at the numbers the trend may seem significant

Our response:

Consistent with all of the similar analyses, we also conducted repeated measure test to examine the effect on LDL. The significance of this analysis (p=.35) was not determine as significant (p<.05) or marginal significant (p<.10).

Comment#5:

A clearer presentation of the baseline vs. pre-lockdown vs. post lock-down characteristics of the study population could be useful: restructuring Table 2 to include trends of variables like body weight, BMI (and BMI >30), METs, etc. would be informative and may identify potential topics which could be amenable of future improvement.

Our response: Data regarding BMI, body weight, body fat percentage, visceral fat level and LDL-C are presented in thabe-2. Data regarding exercise capacity are presented in figure 1. 

Minor Comments:

Comment#1:

 Introduction: The introduction paragraph has too many references (20).

Our response: We removed several references accordingly.

Comment#2:

Methods: the study design has not been reported

Our response:

This was an observational retrospective study. We have added this information to the methods section. (Line 69)

Comment#3:

Methods: a reference for the 10% cut-off to consider capacity improvement could be useful.

Our response: Done. (Reference 20 in the revised manuscript)

Comment#4

 Tables: please add measurement units whenever missing and at least 1 decimal.

Our response: Done.

Comment#5:

Table 1: how many patients were obese?

Our response:

63 patients had a BMI > 30 at baseline. This information has been added to table 1. 

Comment#6:

 Table 1: do the Authors have echocardiographic information?

Our response:

Data regarding the left ventricular systolic function at baseline are presented in table-1.

Comment#7:

 Table 1: PVD abbreviation is for PAD?

Answer#7

Table 1 has been corrected accordingly.

Comment#8:

8. Figure 1: please add number of patients.

Answer#8 :

The number of patients has been added to the figure legend.

[20] Haskiah F, Shacham Y, Minha S, Rozenbaum Z, Pereg D. CHA2DS2-VASc score and exercise capacity of patients with coronary artery disease participating in cardiac rehabilitation programs. Coron Artery Dis. 2017;28(8):697-701.

---

## [Decision Letter · Decision Letter 1]

29 Sep 2022

The Impact of Covid-19 Pandemic on Cardiac Rehabilitation of patients following acute coronary syndrome.

PONE-D-22-14087R1

Dear Dr. Pereg,

We’re pleased to inform you that your manuscript has been judged scientifically suitable for publication and will be formally accepted for publication once it meets all outstanding technical requirements.

Kind regards,

Giuseppe Limongelli

Academic Editor

PLOS ONE

Additional Editor Comments (optional):

Reviewers' comments:

Reviewer's Responses to Questions

**Comments to the Author**

1. If the authors have adequately addressed your comments raised in a previous round of review and you feel that this manuscript is now acceptable for publication, you may indicate that here to bypass the “Comments to the Author” section, enter your conflict of interest statement in the “Confidential to Editor” section, and submit your "Accept" recommendation.

Reviewer #1: All comments have been addressed

Reviewer #2: All comments have been addressed

2. Is the manuscript technically sound, and do the data support the conclusions?

Reviewer #1: Yes

Reviewer #2: Yes

3. Has the statistical analysis been performed appropriately and rigorously? 

Reviewer #1: Yes

Reviewer #2: Yes

4. Have the authors made all data underlying the findings in their manuscript fully available?

Reviewer #1: Yes

Reviewer #2: Yes

5. Is the manuscript presented in an intelligible fashion and written in standard English?

Reviewer #1: Yes

Reviewer #2: Yes

6. Review Comments to the Author

Reviewer #1: The authors have addressed all the comments of the Editors and Reviewers.

I have not further comments.

Reviewer #2: I thank the Authors of ‘The Impact of Covid-19 Pandemic on Cardiac Rehabilitation of patients following acute coronary syndrome’ for acknowledging all the points raised during the reviewing process.

Although the limitations due to the sample size and the difficulty in extrapolating potential factors associated with benefit (or lack thereof) of cardiac rehabilitation during the COVID-19 pandemic may limit generalizability, the results of the report are still of interest.

The Authors may want to report the lack of associated factors with maintained improvement in the Results section and Discussion/Limitations paragraph.

Minor comments.

Please amend all percentages to the first decimal point in figures and tables.

7. PLOS authors have the option to publish the peer review history of their article (what does this mean?). If published, this will include your full peer review and any attached files.

Reviewer #1: No

Reviewer #2: No

---

## [Editor Report · Acceptance letter]

22 Nov 2022

PONE-D-22-14087R1 

The Impact of Covid-19 Pandemic on Cardiac Rehabilitation of patients following acute coronary syndrome. 

Dear Dr. Pereg:

I'm pleased to inform you that your manuscript has been deemed suitable for publication in PLOS ONE. Congratulations! Your manuscript is now with our production department. 

Kind regards, 

on behalf of

Dr. Giuseppe Limongelli 

Academic Editor

PLOS ONE